# Analysis of Genetic Alterations in Tunisian Patients with Lung Adenocarcinoma

**DOI:** 10.3390/cells8060514

**Published:** 2019-05-28

**Authors:** Dhoha Dhieb, Imen Belguith, Laura Capelli, Elisa Chiadini, Matteo Canale, Sara Bravaccini, Ilhem Yangui, Ons Boudawara, Rachid Jlidi, Tahya Boudawara, Daniele Calistri, Leila Ammar Keskes, Paola Ulivi

**Affiliations:** 1Biosciences Laboratory, Istituto Scientifico Romagnolo per lo Studio e la Cura dei Tumori (IRST) IRCCS, 47014 Meldola, Italy; dhoha.dhieb@gmail.com (D.D.); laura.capelli@irst.emr.it (L.C.); elisa.chiadini@irst.emr.it (E.C.); matteo.canale@irst.emr.it (M.C.); sara.bravaccini@irst.emr.it (S.B.); daniele.calistri@irst.emr.it (D.C.); 2Laboratory of Human Molecular Genetics, Faculty of Medicine of Sfax, University of Sfax, Sfax 3029, Tunisia; mayno87.ib@gmail.com (I.B.); ammarkeskesl@gmail.com (L.A.K.); 3Department of Respiratory and Sleep Diseases, CHU Hedi Chaker, Sfax 3029, Tunisia; ilhem.bouazizyangui@gmail.com; 4Department of Pathology, CHU Habib Bourguiba, Sfax 3029, Tunisia; annoussa1988@hotmail.com (O.B.); tahya.sellami@rns.tn (T.B.); 5Laboratory of Anatomic Pathology, Sfax 3000, Tunisia; rachid.jlidi@gmail.com

**Keywords:** molecular profile, EGFR, KRAS, ALK, p53, lung adenocarcinoma

## Abstract

The identification of the mutations that drive lung cancer have furnished new targets for the treatment of non-small cell lung cancer (NSCLC) and led to the development of targeted therapies such as tyrosine kinase inhibitors that are used to combat the molecular changes promoting cancer progression. Furthermore, biomarkers identified from gene analysis can be used to detect early lung cancer, determine patient prognosis, and monitor response to therapy. In the present study we analyzed the molecular profile of seventy-three Tunisian patients with lung adenocarcinoma (LAD). Mutational analyses for *EGFR* and *KRAS* were performed using direct sequencing, immunohistochemistry or MassARRAY. Anaplastic lymphoma kinase (ALK) rearrangement was evaluated by immunohistochemistry using the D5F3 clone, and p53 expression was also assessed. The median age of patients at diagnosis was 61 years (range 23–82 years). Using different methodologies, *EGFR* mutations were found in 5.47% of patients and only exon 19 deletions “E746-A750 del” were detected. *KRAS* mutations were present in 9.58% of cases, while only one patient was ALK-positive. Moreover, abnormal immunostaining of p53 was detected in 56.16% of patients. In conclusion, the detected rates of *EGFR* and *KRAS* mutation and ALK rearrangement were lower than those found in European and Asian countries, whereas, abnormal p53 expression was slightly more frequent. Furthermore, given the small sample size of this study, a more comprehensive analysis of this patient set is warranted.

## 1. Introduction

In the past few years the discovery of driver mutations has led to improvements in targeted therapies for lung adenocarcinoma (LAD) [1]. Epidermal growth factor receptor (EGFR) and anaplastic lymphoma kinase (ALK) alterations are the major actionable genetic alterations that are treatable in LAD [2]. *EGFR* mutations are present in around 15% of patients in Western countries and in 50% of Asians [3,4]. These driver mutations are most frequently found in Asian women and in those who have never smoked [5,6]. They have been shown to be predictive of response to first-line treatment with specific tyrosine kinase inhibitors (TKIs) such as gefitinib, erlotinib, and afatinib [7,8], which have led to improved clinical outcomes as compared with standard chemotherapy [9]. ALK rearrangements, a targetable genetic change occurring in around 5% of patients with LAD [10,11], are more frequent in younger individuals who are light or never smokers [12,13]. The above mutations are particularly sensitive to targeted therapies such as crizotinib, alectinib, and ceritinib [14,15]. Screening for *EGFR* and ALK alterations is now mandatory to determine the appropriate treatment for patients with LAD [4] and should thus be standard practice in the diagnostic workup. In addition, Kirsten rat sarcoma viral oncogene homolog (*KRAS*) testing is now performed more frequently in LAD. Some studies have reported that *KRAS* mutations are more prevalent in women, while others have found no differences with regard to gender [16,17]. Although alterations in *EGFR*, *KRAS*, and ALK are considered to be mutually exclusive [12,18], there is also evidence that these mutations may overlap [19,20]. On the other hand, tumor-suppressor gene 53 (*TP53*), which encodes for a protein (p53) regulating cell-cycle arrest, senescence, and apoptosis, is one of the most commonly altered genes in lung cancer. It can thus be hypothesized that loss of p53 function through mutation leads to unchecked proliferation, tumor growth, and therapeutic resistance [21,22]. *TP53* gene mutations and epigenetic alterations can be detected by immunohistochemistry (IHC) and frequently result in an accumulation of abnormal protein within tumor cells [23]. The incidence of *TP53* mutations in LAD is ~50% according to recent studies and is higher in smokers [24]. International guidelines also recommend that other novel molecular targets such as BRAF, NRAS, PIK3CA, ALK, ERBB2, DDR2, RET, and MAP2K1 variants be analyzed before starting treatment with palliative intent for LAD [25]. The molecular profiling of lung cancer is still a fairly infrequent practice in Tunisia and chemotherapy remains the primary treatment for non-small cell lung cancer (NSCLC). This is in strong contrast to European, Asiatic, and American countries where molecular profiling is well established and different sequencing approaches have been validated [26,27,28]. Evaluating the mutation status of lung cancer patients provides valuable information that can help to identify the optimal treatment regimen for patients. In the present study we used different methodologies to analyze the molecular profile of a representative set of Tunisian patients with LAD. We also underlined the importance of different sequencing approaches (MassARRAY and Sanger) and the usefulness of IHC in the molecular profiling of LAD.

## 2. Materials and Methods

### 2.1. Patients and Tissue Samples

Seventy-three formalin-fixed, paraffin-embedded (FFPE) tissue samples (39 lung biopsies and 34 lung resections) from Tunisian patients with LAD at the Department of Pathology in Habib Bourguiba Hospital in Sfax, Tunisia were analyzed. Hematoxylin- and eosin-stained slides were prepared for each sample and reviewed by 2 experienced pathologists to identify the areas of highest tumor density (i.e., with at least 80% tumor content). Histologic classification was made according to the 2015 WHO criteria [29]. Patients with mixed histology and a coexisting non-pulmonary malignancy were excluded. Clinical and pathological data were obtained from medical records and centrally reviewed for the purposes of the study (Table 1). The overall case series was analyzed using different methodologies (Figure 1).

### 2.2. Molecular Analysis

#### 2.2.1. DNA Extraction

Macrodissection was performed using the tip of a blade to scrape off the selected tumor areas on 5-µm unstained slides based on the tumor area selected in hematoxylin- and eosin-stained slides. Cells were lysed in 50 mM of KCl, 10 mM of Tris-HCl pH 8.0, 2.5 mM of MgCl_2_ and Tween-20 0.45% supplemented by proteinase K at a concentration of 1.25 mg/mL, overnight at 56 °C. Proteinase K was inactivated at 95 °C for 10 min, after which samples were centrifuged twice to eliminate debris. DNA was purified using QIAamp DNA micro kit (Qiagen, Hilden, Germany) following the cleanup of genomic DNA protocol. After assessing the quantity and quality of the DNA by Nanodrop-2000 (Thermo Fisher Scientific, Monza, Italy), it was stored at −20 °C for further testing.

#### 2.2.2. Mutation Analysis

Exon 2 of *KRAS* and exons 18–21 of *EGFR* were amplified in 53 of the 73 cases by polymerase chain reaction (PCR) assay using the primers indicated in Table 2. PCR product was purified using exonuclease and then submitted to sequencing using an ABI PRISM 3100-Avant automated DNA sequencer with the BigDye Terminator Cycle Sequencing reaction kit v1.1. (catalog number 4337450, Thermo Fisher Scientific, Monza, Italy). Each exon was sequenced on both strands, amplified, and then sequenced again on both strands to eliminate the risk of PCR artifacts. *KRAS* and *EGFR*, together with *BRAF*, *NRAS*, *PIK3CA*, *ALK*, *ERBB2*, *DDR2*, *RET*, and *MAP2K1* gene mutations were also analyzed by MassARRAY platform (Sequenom, San Diego, CA, USA) using Myriapod Lung Status CE-IVD kit (Diatech Pharmacogenetics, Jesi, Italy) according to the manufacturer’s instructions. The MassARRAY (Sequenom) testing was carried out on 20 cases, and the selection of the methodology to be used was done on the basis of the amount of tumor tissue available.

### 2.3. Immunohistochemistry

#### 2.3.1. EGFR Mutation-Specific Immunohistochemistry

Immunohistochemical staining was performed on 24 cases. We used EGFR E746-A750del (catalog number 2085, Cell Signaling Technologies (CST), Danvers, MA, USA) and EGFR L858R (catalog number 3197, Cell Signaling Technologies (CST)) as primary antibodies that were manually applied to the slides. The technique was carried out on a Benchmark^®^ GX (Ventana Medical Systems, Tucson, AZ, USA) automated stainer. Immunoreactivity was revealed with UltraView Universal DAB detection kit (Ventana Medical Systems) and slides were counterstained with hematoxylin. Positive and negative controls were used. Each slide was examined and scored independently by two pathologists (Table 3) [30]. A specimen was considered positive in the presence of nuclear and/or cytoplasmic immunostaining for EGFR in tumor cells. Any discordance was resolved by consensus after a joint review using a multihead microscope.

#### 2.3.2. ALK Expression Analysis

Immunohistochemistry was performed on all cases with a monoclonal rabbit ALK antibody (D5F3, Ventana Medical Systems) using an OptiView DAB IHC Detection Kit and an OptiView Amplification Kit. The technique was fully automated on Benchmark GX according to the manufacturer’s recommendations. A sample was classified as positive if strong granular cytoplasmic staining was present in any percentage of tumor cells and negative if the tumor did not show immunoreactivity or if there was only weak or moderate cytoplasmic staining.

#### 2.3.3. p53 Expression Analysis

Immunohistochemical staining was performed using p53 monoclonal antibodies (Clone: DO-7, code: M7001, dilution, 1/50, Dako). Positive and negative controls were used to validate the reactions. p53 immunostaining was assessed semiquantitatively on the basis of staining intensity, and proportion of positive tumor cells (Table 4).

### 2.4. Compliance with Ethical Standards and Informed Consent

The study was conducted in accordance with the Declaration of Helsinki and was approved by the Ethics Committee of Sfax University, Tunisia, (CPP SUD N° 34/2016, 10/2016). The need for informed consent was waived because of the retrospective nature of the study.

## 3. Results

### 3.1. Patient Characteristics

Of the 73 patients with LAD enrolled in the present study, 12 (16.43%) were females and 61 (83.56%) were males. The median age was 61 years (range 23–82 years). The FFPE samples were adequate for final diagnosis and molecular analysis in all cases. The TNM classification was available for 53 patients, 33 (62.26%) classified with stage III or IV disease and 20 (37.73%) with stage I or II, and 36.53% (19/52) of patients had metastases. Smoking status was known for 59 patients of whom 45 patients were smokers (76.27%) and 14 non-smokers. Full patient characteristics are shown in Table 1.

### 3.2. EGFR and KRAS Analysis

For the *EGFR* and *KRAS* analysis, 53 patients were studied by PCR followed by Sanger sequencing and the remaining 20 were evaluated by MassARRAY (Sequenom) (Figure 1). The EGFR E746-A750 del and EGFR L858R expression was assessed by IHC in 24 patients (Figure 1). Of the 24 patients analyzed by IHC (Table 3), only one (4.16%) harbored an EGFR mutation (E746-A750 del) (Table 5). Abnormal immunolabeling of EGFR was detected in a 70-year old non-smoker male with acinar histological subtype (Figure 2 and Table 3). Cells showing membranous/cytoplasmic staining alone or in association were considered positive and thus scored. None of the patients analyzed by EGFR mutation-specific IHC harbored an L858R mutation. *EGFR* mutations were also detected by MassARRAY in three (15%) other patients (two males, one smoker and one non-smoker, and one non-smoker female) (Figure 3, Table 5). Four (5.47%) of the 73 patients harbored an *EGFR* mutation and the E746-A750 del of exon 19 (2235–2249 del) was only detected in three (4.1%) non-smokers (one female and two males) and in one (1.36%) male smoker. A comparison between wild-type and mutant *EGFR* patients with regard to gender revealed that 1.36% of females and 4.1% of males had an *EGFR* mutation. In terms of smoking status, 4.1% of non-smokers and only 1.36% of smokers showed an *EGFR* mutation (Table 5 and Table 6). Conversely, *KRAS* mutations were detected in seven (9.58%) patients of whom six (8.21%) were male (all smokers) and one (1.36%) was female (non-smoker). Three of the *KRAS*-mutated patients were identified using the MassARRAY (Figure 4A,B) and the remaining four patients were identified using Sanger sequencing (Figure 5). Two of the seven patients showed a G12D mutation, three patients a G13D mutation, one patient a G12A mutation, and one patient a G12V mutation (Table 5 and Table 6).

### 3.3. ALK and p53 Protein Expression Analysis

Of the 73 patients, only one (77-year-old smoker) showed strong granular, homogenous and diffuse ALK cytoplasmic staining (Figure 6 and Table 5). With regard to p53, only nuclear immunostaining was considered. Tumors showing immunoreactivity of >10% of tumor cell nuclei were considered positive (p53 overexpression). Abnormal p53 immunostaining was detected in 41 (56.16%) cases and 21 (28.76%) showed an overexpression of p53 protein (Figure 7).

## 4. Discussion

In the present study we performed a preliminary analysis of the molecular profile of Tunisian patients with lung adenocarcinoma, using different technologies. Our results highlighted the importance of different sequencing approaches (MassARRAY and Sanger sequencing) and the usefulness of IHC in helping to select the best therapeutic strategy. Our investigation revealed that the profile of the different mutations was fairly similar to that of European populations, albeit with a lower prevalence.

The most common *EGFR* mutations associated with NSCLC (E746-A750 del and L858R), which together accounted for 86–90% of the total number of *EGFR* mutations (45% for E746-A750 del and 40–45% for L858R) [31,32,33], were investigated in 24 patients using IHC (Table 1). Abnormal immunolabeling of EGFR was detected in only one (4.16%) case, in contrast to the much higher incidence (44%) recently reported by Mraihi et al. in 50 Tunisian patients [34]. In this context, doing a molecular verification by Sanger sequencing, after performing IHC, would be ideal for the positive case. However, the small size of the biopsy did not allow us to make an additional test. We also used MassARRAY (Sequenom) technology to study the molecular profile of 20 cases, detecting *EGFR* mutations in three (15%) patients, all of whom harbored the E746-A750 del of exon 19. This incidence was similar to that found in European countries [35]. Furthermore, our results were in agreement with data from a recent Tunisian study by Arfaoui et al. on *EGFR* mutations determined by Therascreen *EGFR* RGQ PCR kit (Qiagen) [36] where the authors reported an incidence of 11.5% (3/26 cases) in LAD patients. Conversely, we did not detect any *EGFR* mutations using Sanger sequencing, which could be attributed to the low sensitivity and need for high-quality tumor samples of this technology [37]. Our results prove that the variability in frequency was primarily related to the sensitivity of the method used to analyze the markers. In general, mutation averages vary from study to study and from country to country, and this variability is probably related to patient selection criteria (clinical and pathological) and to the methods used for mutational analysis. We also observed that *EGFR* mutations were more frequent in males and non-smokers than in females and smokers (Table 5 and Table 6). However, it has been seen that higher *EGFR* mutation frequencies were observed in female non-smokers of Asian origin [38,39], reaching more than 60% in this population.

Using both Sanger sequencing and Sequenom technology in the second part of our investigation, we observed a similar incidence of *KRAS* mutations in LAD patients with respect to that previously observed in an Asian population [40]. The frequency was lower than that seen in Caucasian populations (around 30%) [41,42], which may have been due to the smoking habits of the group studied. A recent Chinese study on smokers with LAD reported a 14.0% incidence of *KRAS* mutations, while that of LAD non-smokers was 3.4% [43]. In our study, *KRAS* mutations were observed more frequently in male smokers than in female non-smokers (Table 5 and Table 6). With regard to ALK expression, we detected positivity in only one (1.36%) of the 73 patients using IHC. Interestingly, our IHC results on ALK differ from those of other studies, independently of the method used. In fact, ALK rearrangements frequencies of 3–5% were observed in unselected patient populations [44,45,46] and of 33% in highly selected patient populations (*EGFR* wild type, female, non/light smokers) [47,48]. ALK rearrangement was usually associated with young age, female gender, and a no- or light-smoking history [47,49]. A Tunisian study by Mezni et al. [50] showed a prevalence of ALK rearrangement in six (7.14%) out of 84 patients using IHC, which differs fairly substantially from our findings. Furthermore, Liang et al. [51] suggested that the incidence of ALK rearrangement in the Chinese population may be correlated with smoking status as they observed a lower prevalence in smokers (2.9%) than in non-smokers (7.2%). Smoking and a higher percentage of males in our study may account for the lower incidence of ALK translocations detected. With regard to p53, abnormal immunolabeling was detected in 41 cases (56.16%). There have been some reports of gender differences in p53 mutations [52], and several studies have indicated that *TP53* mutations occur more frequently in smokers than in non-smokers [52,53]. In our group, abnormal p53 expression was most frequently observed in males and smokers. However, we did not find a significant correlation between gender or tobacco consumption and p53 mutation.

In summary, our preliminary study highlighted the importance of different sequencing approaches (MassARRAY and Sanger sequencing) and the usefulness of IHC in helping to select the best therapeutic strategy. Furthermore, our investigation revealed that the profile of the different mutations was fairly similar to that of European populations, albeit with a lower prevalence for EGFR, ALK, and KRAS and slightly higher for p53 expression. In addition, our finding encourages the use of the Sequenom as an easy specific and most sensitive approach for the screening of LAD patients, however, this brought us to wonder about the cost especially in low income countries. Further studies with larger Tunisian series are required to obtain more conclusive results.

## Figures and Tables

**Figure 1 cells-08-00514-f001:**
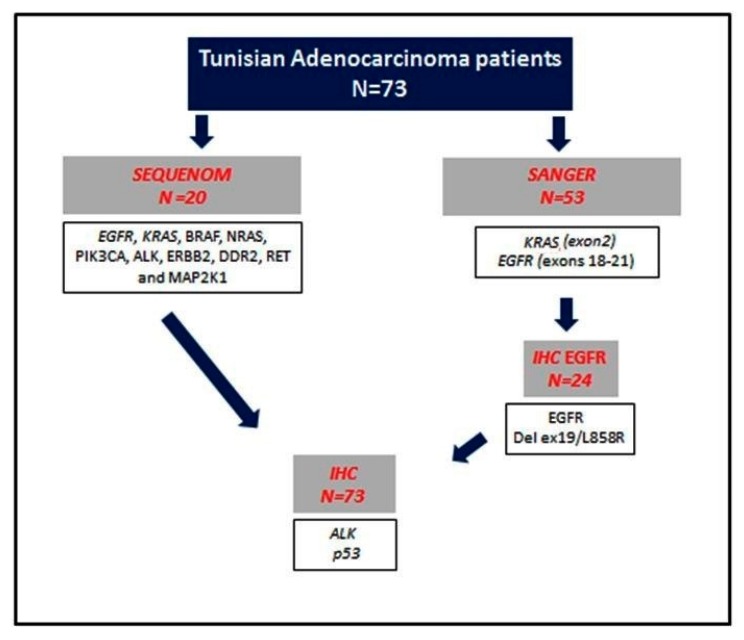
Flow chart of the analytical methods used.

**Figure 2 cells-08-00514-f002:**
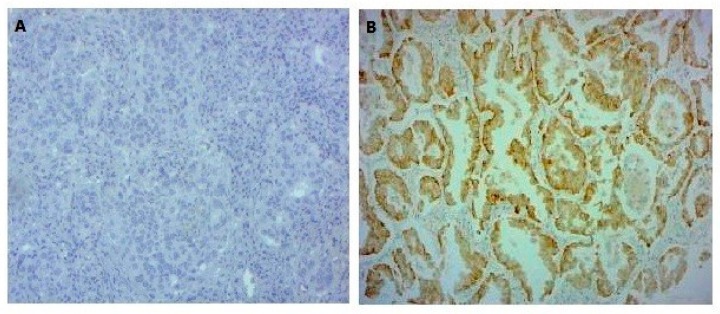
EGFR immunohistochemistry (IHC) staining in lung adenocarcinoma: (**A**) Negative immunostaining of EGFR (200×), (**B**) EGFR cytoplasmic IHC staining in lung adenocarcinoma (LAD). Strong positivity of EGFR in cytoplasm [3+, 100%] (200×).

**Figure 3 cells-08-00514-f003:**
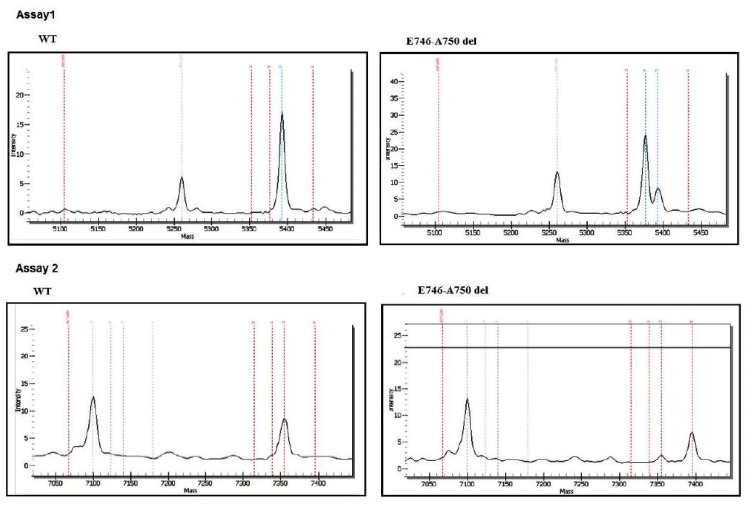
Spectrum of the Sequenom assay shows *EGFR* E746-A750 del (exon 19).

**Figure 4 cells-08-00514-f004:**
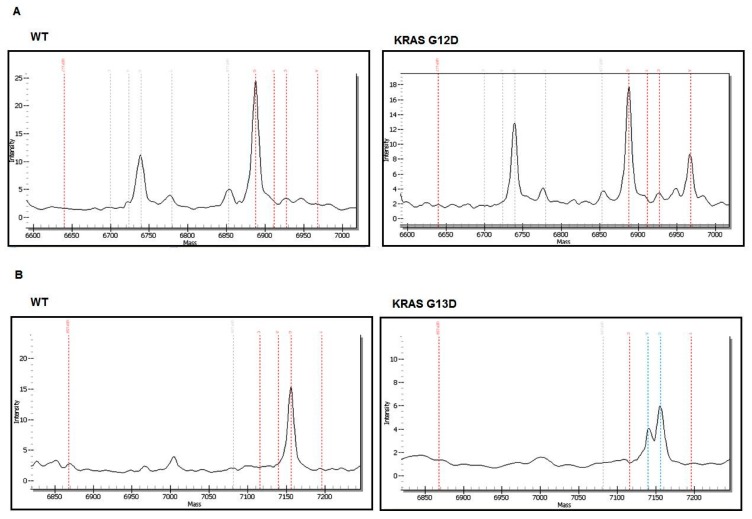
Spectrum of the Sequenom assay shows *KRAS* mutations. (**A**) G12D *KRAS* mutation (exon 2 codon 12), (**B**) G13D *KRAS* mutation (exon 2 codon 13).

**Figure 5 cells-08-00514-f005:**
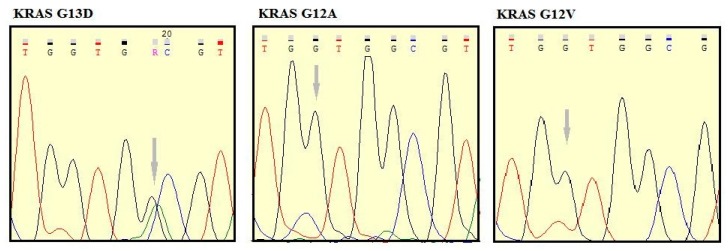
Sequencing by the standard method shows a mutant peak consistent with a G13D, G12A, and G12V mutation.

**Figure 6 cells-08-00514-f006:**
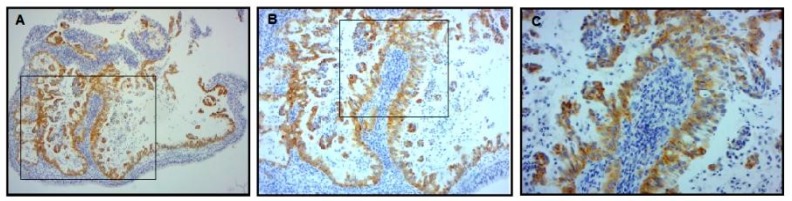
Anaplastic lymphoma kinase (ALK) protein expression: Tumor harboring positive ALK expression shows strong granular and homogenous cytoplasmic staining, (**A**) 100×, (**B**) 250×, (**C**) 400×.

**Figure 7 cells-08-00514-f007:**
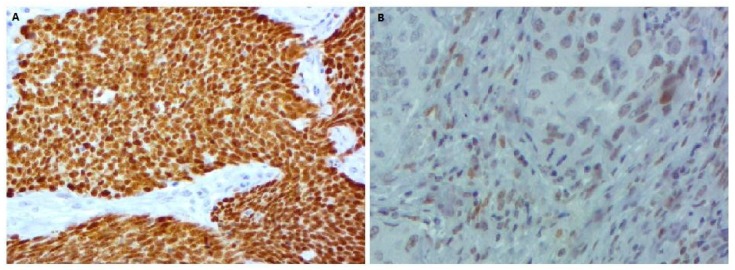
P53 protein expression in lung adenocarcinoma: (**A**) strong immunostaining (p53 score = 12), (**B**) low immunostaining (p53 score = 4) (400×).

**Table 1 cells-08-00514-t001:** Clinical pathological characteristics of patients (*n* = 73).

	No. (%)
All patients	73 (100)
Age, years	73
≤61	36 (49.31)
>61	37 (50.68)
Gender	73
Male	61 (83.56)
Female	12 (16.43)
Tumor stage	53
I–II	20 (37.73)
III–IV	33 (62.26)
Tumor size	44
T1–T2	21 (47.72)
T3–T4	23 (52.27)
Lymph node (N) involvement	44
N0–N1	27 (61.36)
N2–N3	17 (38.63)
Metastasis	52
M0	33 (63.46)
M1	19 (36.53)
Smoking status	59
Non-smoker	14 (23.72)
Smokers	45 (76.27)

**Table 2 cells-08-00514-t002:** Primer sequences.

	Forward Primer	Reverse Primer	Annealing Temperature (°C)	Product Size (bp)
*KRAS* exon 2	GACTGAATATAAACTTGTGGTAGTTGG	TTGGATCATATTCGTCCACAA	60	101
*EGFR* exon 18	TCCAAATGAGCTGGCAAGTG	TCCCAAACACTCAGTGAAACAAA	60	397
*EGFR* exon 19	CGTCTTCCTTCTCTCTCTGTC	GACATGAGAAAAGGTGGGC	60	190
*EGFR* exon 20	CATTCATGCGTCTTCACCTG	CATATCCCCATGGCAAACTC	60	377
*EGFR* exon 21	GCTCAGAGCCTGGCATGAA	CATCCTCCCCTGCATGTGT	60	348

**Table 3 cells-08-00514-t003:** Epidermal growth factor receptor (EGFR) immunostaining score.

Staining Intensity	No Staining 0	Weak 1	Moderate 2	Strong 3	Very Strong 4
Proportion of positive tumor cells	0% to 100%				
	Immunostaining score = positive cell proportion score multiplied by staining intensity score H = 1 × (% cells 1+) + 2 × (% cells 2+) + 3 × (% cells 3+) + 4 × (% cells 4+)				
	0–200			200–400	
Degree of immunostaining	*Negative or low*			*High*	

**Table 4 cells-08-00514-t004:** p53 immunostaining score.

Score	0	1	2	3	4
Staining intensity	No staining	Light yellow	Yellowish brown	Brown	Dark brown
Proportion of positive tumor cells	0%	<10%	11% to 50%	51% to 80%	>80%
	Immunostaining score = positive cell proportion score multiplied by staining intensity score				
Degree of immunostaining	0		1–4	>4	
	Negative		Low	High	

**Table 5 cells-08-00514-t005:** Clinical pathological data and *EGFR*, *KRAS*, and *ALK* status of patients with genetic alterations.

Patient No.	Gender	Age, Years	Smoking Data	Specimens	pTNM/Stage Grouping	Histological Subtype	Genetic Alteration	Method
1	M	60	Yes	Biopsy		Acinar	*EGFR*: EX-19 pE746-A750del	Sequenom
2	M	53	Yes	Biopsy		Acinar	*KRAS* G13D	Sequenom
3	F	61	No	Surgical	pT2bN1M0/II	Mucinar	*KRAS* G12D	Sequenom
4	M	74	Yes	Biopsy		Lepidic	*KRAS* G12D	Sequenom
5	F	53	No	Biopsy		Solid	*EGFR*: EX19 pE746-A750del	Sequenom
7	M	70	No	Biopsy		Acinar	*EGFR*: EX19 pE746-A750del	IHC
8	M	71	No	Surgical	pT1N1M0/II	-	*EGFR*: EX19 pE746-A750del	Sequenom
9	M	61	Yes	Biopsy		Lepedic	*KRAS* G12A	Sanger
10	M	66	Yes	Biopsy		-	*KRAS* G13D	Sanger
11	M	58	Yes	Biopsy		Solid	*KRAS* G12V	Sanger
12	M	62	Yes	Biopsy		Acinar	*KRAS* G13D	Sanger
13	M	77	Yes	Biopsy		-	ALK +	IHC

IHC, immunohistochemistry.

**Table 6 cells-08-00514-t006:** Genetic alteration in relation to gender and the smoking history of patients with LAD (*n* = 73).

	EGFR+	KRAS+	ALK+	Abnormal p53 Expression
Gender				
Male	3	6	1	37
Female	1	1	0	4
Smoking history				
Non-smoker	3	1	0	5
Smoker	1	6	1	23

LAD, lung adenocarcinoma.

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
