# Peer review of "Analysis of Genetic Alterations in Tunisian Patients with Lung Adenocarcinoma"

_cells, 2019, doi:10.3390/cells8060514_

Reviewer 1 Report

In this manuscript, Dhieb et al. characterize the contribution of the major drivers of lung adenocarcinoma in Tunisian patients. As such, the authors evaluate EGFR, KRAS, ALK fusions and TP53 status in all tumors and BRAF, NRAS, PIK3CA, ERBB2, DDR2, RET, MEK1, and ALK mutations in a subset of tumors and found that EGFR, KRAS, and ALK were altered less frequently than has been documented in cohorts of European or Asian populations. The authors also find that TP53 appears to be altered more frequently in lung tumors from Tunisian patients. In principle, this is an important study, as it helps clarify the similarities and points to the potential differences in the genesis of lung adenocarcinoma among different populations.   However, the conclusions reached by the authors most likely reflect issues in detection methodology and do not definitely demonstrate that there is variance in the Tunisian cohort compared to those that have been characterized more extensively.  The authors would need to state this explicitly or re-address the mutational frequencies using Sequenom or other technologies before claiming significant differences in KRAS, EGFR, ALK and TP53 mutation prevalence. Minor changes/additions would add to the clarity of the manuscript and one ‘major’ change (re: KRAS mutation detection and EGFR mutation detection) is highlighted below as it relates to the appropriate section or figure. Introduction -On lines 46-48 the authors imply that there are approved treatments for tumors with KRAS alterations (“actionable genetic alterations”). There are none. -On line 68, the authors state: “The incidence of TP53 mutations in LAD is ~50% according to recent studies and is higher in smokers [24].” This is fairly similar to what the authors find in the Tunisian cohort so the abstract should be modified to state this as a similarity rather than a stark difference. Results -Did the one patient that stained positive with the anti-EGFR deletion antibody also score as a deletion mutant by Sanger sequencing? This should be stated in the text. -Can the authors comment on why some samples were processed by Sequenom vs. others by Sanger sequencing? Was this part of the study design? -How well does staining with the DF3 antibody capture the spectrum of ALK-fusions and/or ALK mutations? - Are the panels on the right side of Figure 3 flipped? -In the middle and right chromatograms in Figure 5, the ‘C’ peak is very low (GCT, Alanine) as is the ‘T’ peak (GTT, Valine). A major claim of this paper is that the mutation frequency in exon2 of KRAS seem to be low, under-represented. However, the Sanger sequencing assay, as presented, does not appear to robustly capture the mutations. The authors should clone (TA-cloning) the PCR products and sequence those products (assuming the ~80% tumor purity the authors note, a mutation should be captured at least once as a single peak from ~20 clones) to unambiguously determine if a mutation in KRAS is present. Or, the 53 samples that were analyzed by PCR should all be analyzed by Sequenom technology or other methodology. (https://www.ncbi.nlm.nih.gov/pubmed/28618430) Discussion: -The authors mention that their cohort contained cases with fewer mutations in EGFR than detected by another group that also examined Tunisian patients (by IHC). The authors speculate that this might be due to the sensitivity of methodologies (Sanger sequencing): “Our results proved that the variability in frequency was primarily related to the sensitivity of the method used to analyze the markers. “   -This statement to explain the confounding results is problematic. 24 of the 73 samples were analyzed by IHC for EGFR mutations. The paper, as articulated in the abstract says: “In conclusion, the detected rates of EGFR and KRAS mutation and ALK rearrangement were lower than those found in European and Asian countries, whereas abnormal p53 expression was more frequent.” -However, if this is an issue of detection sensitivity, the authors are actually saying that Sanger sequencing—as performed in this paper--- should not be used for mutation status determination. -The authors also mention that another study found a higher incidence of ALK mutations in a Tunisian population; a frequency more similar to that reported in European populations (ref. 50).  This draws into question the reliability of the DF3 antibody to detect ALK fusions and mutations. -The authors state: “In conclusion, it is difficult to determine the real prevalence of mutations in lung adenocarcinoma as many selection biases occur.” This statement is misleading. The difficulty lies in detection methodology and its associated costs, however, the real prevalence of common, truncal mutations can be reliably determined.  Definitely knowing the mutation status of tumors forms the basis for many present and all future tumor therapies.

Author Response

REVIEWER#1

1#.       Introduction -On lines 46-48 the authors imply that there are approved treatments for tumors with KRAS alterations (“actionable genetic alterations”). There are none. 

Our response / action taken: We thank the referee for this observation, and we have corrected the sentence.

2#.       On line 68, the author’s state: “The incidence of TP53 mutations in LAD is ~50% according to recent studies and is higher in smokers [24].” This is fairly similar to what the authors find in the Tunisian cohort so the abstract should be modified to state this as a similarity rather than a stark difference.

Our response / action taken: This comment was also taken into account and the term “more frequent” has been replaced by “slightly more frequent” (line 37).

3#.       Did the one patient that stained positive with the anti-EGFR deletion antibody also score as a deletion mutant by Sanger sequencing? This should be stated in the text.

Our response / action taken: Your significant question has been answered as fellow:  We studied epidermal growth factor receptor (EGFR) expression profile for patients with LAD from whom tumor materials are not sufficient for molecular investigations (including the one patient #7 that stained positive) and therefore, using IHC is a better management of small size and/or low content tumor cells samples which not always making molecular studies possible. Besides, our work was retrospective, and we don’t have the right to exhaust the specimens of patients. Table 5 provides more information about the method used for the detection of the genetic alteration in each case. This note is added to the discussion section (line361-363) as you had recommended, and we hope will meet your expectations.

4#.       Can the authors comment on why some samples were processed by Sequenom vs. others by Sanger sequencing? Was this part of the study design?

Our response / action taken: We acknowledge this comment raised by the reviewer. In addition to what we mention above, the small size of the collected samples in addition to the quality of the DNA and the sensitivity of the method used to analyze the markers, prompted us to choose between Sequenom and Sanger sequencing.

5#.       How well does staining with the DF3 antibody capture the spectrum of ALK-fusions and/or ALK mutations?

Our response / action taken: According to many reports, D5F3 clone have shown a strong reliability in the detection of ALK-fusions and/or ALK mutations, thus could generated a high sensitivity of 100% and specificity that reach 99% comparing to other clones. In addition, recent studies showed that immunohistochemistry with D5F3 antibody could correlate equally well with FISH to detect ALK rearrangements. The following references could provide more details.

1.        Wynes, M.W.; Sholl, L.M.; Dietel, M.; Schuuring, E.; Tsao, M.S.; Yatabe, Y.; Tubbs, R.R.; Hirsch, F.R. An international interpretation study using the ALK IHC antibody D5F3 and a sensitive detection kit demonstrates high concordance between ALK IHC and ALK FISH and between evaluators. J. Thorac. Oncol. 2014.

2.        Yi, E.S.; Boland, J.M.; Maleszewski, J.J.; Roden, A.C.; Oliveira, A.M.; Aubry, M.C.; Erickson-Johnson, M.R.; Caron, B.L.; Li, Y.; Tang, H.; et al. Correlation of IHC and FISH for ALK gene rearrangement in non-small cell lung carcinoma: IHC score algorithm for FISH. J. Thorac. Oncol. 2011.

3.        Taheri, D.; Zahavi, D.J.; Del Carmen Rodriguez, M.; Meliti, A.; Rezaee, N.; Yonescu, R.; Ricardo, B.F.P.; Dolatkhah, S.; Ning, Y.; Bishop, J.A.; et al. For staining of ALK protein, the novel D5F3 antibody demonstrates superior overall performance in terms of intensity and extent of staining in comparison to the currently used ALK1 antibody. Virchows Arch. 2016.

6#.       Are the panels on the right side of Figure 3 flipped?

Our response / action taken: we thank the referee’s for this observation. This was a mistake and the figure has been corrected.

7#.       In the middle and right chromatograms in Figure 5, the ‘C’ peak is very low (GCT, Alanine) as is the ‘T’ peak (GTT, Valine). A major claim of this paper is that the mutation frequency in exon2 of KRAS seem to be low, under-represented. However, the Sanger sequencing assay, as presented, does not appear to robustly capture the mutations. The authors should clone (TA-cloning) the PCR products and sequence those products (assuming the ~80% tumor purity the authors note, a mutation should be captured at least once as a single peak from ~20 clones) to unambiguously determine if a mutation in KRAS is present. Or, the 53 samples that were analyzed by PCR should all be analyzed by Sequenom technology or other methodology. (https://www.ncbi.nlm.nih.gov/pubmed/28618430)

Our response / action taken: We understand the referee’s perplexity about the low peak of the mutated alleles. However, for both cases, sequencing was performed with both forward and reverse primers, confirming in both cases the presence of the mutated peak. Moreover, the lowering of the WT allele peak confirms us the presence of the mutation.

8#.       Discussion: -The authors mention that their cohort contained cases with fewer mutations in EGFR than detected by another group that also examined Tunisian patients (by IHC). The authors speculate that this might be due to the sensitivity of methodologies (Sanger sequencing): “Our results proved that the variability in frequency was primarily related to the sensitivity of the method used to analyze the markers. “This statement to explain the confounding results is problematic. 24 of the 73 samples were analyzed by IHC for EGFR mutations. The paper, as articulated in the abstract says: “In conclusion, the detected rates of EGFR and KRAS mutation and ALK rearrangement were lower than those found in European and Asian countries, whereas abnormal p53 expression was more frequent.” However, if this is an issue of detection sensitivity, the authors are actually saying that Sanger sequencing—as performed in this paper--- should not be used for mutation status determination.

Our response / action taken: The reviewer is perfectly right to raise this comment. Accordingly, the conclusion has been reviewed and updated and necessary changes have been introduced to make it more informative and consistent with the discussion. For this purpose, we indicate that “our preliminary study highlighted the importance of different sequencing approaches (Mass-array and Sanger sequencing) and the usefulness of IHC in helping to select the best therapeutic strategy. Furthermore, our investigation revealed that the profile of the different mutations was fairly similar to that of European populations, albeit with a lower prevalence for EGFR, ALK and KRAS and slightly higher for p53 expression. Besides, our finding encourages the use of the Sequenom as an easy specific and most sensitive approach for the screening of LAD patients, however, this brought us to wonder about the cost especially in low income countries. Further studies with larger Tunisian series are required to obtain more conclusive results”. For more details and illustrations, we would like to refer to the conclusion section (lines 401-408).

9#.       The authors also mention that another study found a higher incidence of ALK mutations in a Tunisian population; a frequency more similar to that reported in European populations (ref. 50). This draws into question the reliability of the DF3 antibody to detect ALK fusions and mutations.

Our response / action taken: We acknowledge the reviewer when he makes approximately the same comment number 5#. In addition to that, ALK positive expression, according to many reports, is well linked with young age, female gender and light or no smoking history. On the other hand, the difference in the frequency found in our study and the reported one by Mezni et al in 2018, do not reflect the reliability of the D5F3 antibody. Moreover, the variability can be related to the clinic-pathological parameters of the studied cases like Age, Gender and smoking history.

10#.     The authors state: “In conclusion, it is difficult to determine the real prevalence of mutations in lung adenocarcinoma as many selection biases occur.” This statement is misleading. The difficulty lies in detection methodology and its associated costs, however, the real prevalence of common, truncal mutations can be reliably determined. Definitely knowing the mutation status of tumors forms the basis for many present and all future tumor therapies. 

Our response / action taken: We understand and agree the reviewer’s comment. In fact, the entire paragraph has been revised and updated as we have considered a new conclusion regarding the difficulties that lies in detection methodology and costs especially in low income countries. We hope this revised version will meet your standards and expectations.

Reviewer 2 Report

In the manuscript “Analysis of Genetic Alterations in Tunisian Patients with Lung Adenocarcinoma” Dhieb et al. examined the molecular profiles of formalin-fixed, paraffin-embedded tissue samples from 73 Tunisian LAD patients. The authors focused on a few currently actionable genetic targets in LAD: KRAS, EGFR and ALK, in addition to the known most commonly affected gene TP53 in lung cancer. Such study provides valuable information on LAD molecular profiling, due to the very limited data currently available in Tunisia.

Major concerns:

1.     The authors used different methods to analyze the different subsets of the 73 patient samples. It is not very clear to the readers the purpose of using different methods. Specifically, there was lack of systematic comparison between the results obtained using different methods. There is lack of conclusion/guidance reached for future molecular diagnosis, do the authors recommend one of the methods over others for each of the mutation detection or the combination of all methods are preferred?

2.     There is lack of explanation on why the 53 samples were selected for SANGER sequencing, while the other 20 samples for SEQUENOM. Given the more consistent sensitivity of IHC on paraffin-embedded samples, in comparison with DNA extraction from FFPE samples, it would make sense to perform IHC on all the samples instead of cherry picking 24 samples only. It would make sense to perform those methods side by side on the same samples to directly compare the results for consistency.

3.     It is not explained why sequencing was not applied to confirm the P53 abnormality detected in 56.16% of the 73 patients. Since this number/percentage is higher than those obtained from other populations. In comparison to the frequency of TP53 mutation detected (3 cases out of 84 patient samples) in another study Tunisian patients by Menzi et al., this is a lot higher.

4.     Given the limited sample size of this study and the discrepancy between the results observed in the current study and the other two studies on Tunisian lung cancer patients, it is recommended that the authors be cautious when drawing conclusion “In conclusion, the 34 detected rates of EGFR and KRAS mutation and ALK rearrangement were lower than those found in 35 European and Asian countries, whereas abnormal p53 expression was more frequent” (in both abstract and conclusion sections), as if the 73 patients representing the whole Tunisian population. In addition, the authors’ conclusion “The present study is the first to 401 analyze the molecular profile of Tunisian patients with lung adenocarcinoma using different technologies” (line 401-402) is not accurate since Mraihi et al. had compared IHC vs sequencing on EGFR mutation side by side in 50 Tunisian lung cancer patients.

Minor concerns:

1.     It is unusual for the authors to use gmail as contacts instead of institutional addresses.

2.     Please be aware of the using of specific epidemiological/statistical terms: recommend to use another word instead of “cohort” in Introduction section line 77 since this is a retrospective study and cross-sectional analysis on pathological samples. Recommend to use median instead of mean in abstract section line 30.

Author Response

REVIEWER#2

Major points:

1#.       The authors used different methods to analyze the different subsets of the 73 patient samples. It is not very clear to the readers the purpose of using different methods. Specifically, there was lack of systematic comparison between the results obtained using different methods. There is lack of conclusion/guidance reached for future molecular diagnosis, do the authors recommend one of the methods over others for each of the mutation detection or the combination of all methods are preferred?

Our response / action taken: The reviewer is perfectly right to raise this comment. Accordingly, the entire conclusion has been re-written and corrected as we have considered your suggestions (Please refer to lines 402-409).

 2#.       There is lack of explanation on why the 53 samples were selected for SANGER sequencing, while the other 20 samples for SEQUENOM. Given the more consistent sensitivity of IHC on paraffin-embedded samples, in comparison with DNA extraction from FFPE samples, it would make sense to perform IHC on all the samples instead of cherry picking 24 samples only. It would make sense to perform those methods side by side on the same samples to directly compare the results for consistency.

 Our response / action taken: We do agree with the reviewer when he makes approximately the same comment raised by reviewer#1. As part of the study design, IHC was performed for patients with LAD from whom tumor materials are not sufficient for molecular investigations. Therefore, IHC is a better management of small size and/or low content tumor cells samples which not always making molecular studies possible, however, IHC is not always possible for the screening of some genes that are necessary for therapeutic decision. Besides, the small amount of tissue does not allow us to perform side by side different methods for the same simple. In this context, selection was made according to the quality of the DNA and the sensitivity of the method used to analyze the markers. In addition, our work was retrospective, and we don’t have the right to exhaust the specimens of patients.

3#.       It is not explained why sequencing was not applied to confirm the P53 abnormality detected in 56.16% of the 73 patients. Since this number/percentage is higher than those obtained from other populations. In comparison to the frequency of TP53 mutation detected (3 cases out of 84 patient samples) in another study Tunisian patients by Mezni et al., this is a lot higher.

Our response / action taken: We largely agree with the reviewer that doing Sanger Sequencing would be ideal to confirm the P53 expression abnormality. However, after performing IHC, it is quite difficult to assess the screening of the whole hot spot of P53 gene (Exon 5 to Exon 9) due to the small amount of the remaining FFPE tissue. However, according to a recent report performed by Mezni et al in 2018, the screening of 41 patients from a total of 84 studied were analyzed by Next Generation Sequencing. Mutations of P53 gene was detected only in 3 cases. So, it is quite difficult to compare the results presented by Mezni et al with our results, especially that many factors would explain this difference in term of specificity of the used methods (mutation/expression) and also the clinic-pathological parameters of the studied cases.

 4#.       Given the limited sample size of this study and the discrepancy between the results observed in the current study and the other two studies on Tunisian lung cancer patients, it is recommended that the authors be cautious when drawing conclusion “In conclusion, the 34 detected rates of EGFR and KRAS mutation and ALK rearrangement were lower than those found in 35 European and Asian countries, whereas abnormal p53 expression was more frequent” (in both abstract and conclusion sections), as if the 73 patients representing the whole Tunisian population. In addition, the authors’ conclusion “The present study is the first to 401 analyze the molecular profile of Tunisian patients with lung adenocarcinoma using different technologies” (line 401-402) is not accurate since Mraihi et al. had compared IHC vs sequencing on EGFR mutation side by side in 50 Tunisian lung cancer patients.

Our response / action taken: As suggested, the conclusion section have been reviewed and necessary changes have been introduced to make it more informative. In addition, according to the new study performed by Mraihi et al. the analysis of 50 Tunisian lung cancer patients showed 44% had EGFR mutation by IHC validated by molecular analyses using Sanger Sequencing, in which we could not talk about molecular profiling especially that only one gene was analyzed detecting only the E746-A750 deletion and L858R mutations.

Other minor point is: 

1#.       It is unusual for the authors to use Gmail as contacts instead of institutional addresses

Our response / action taken: Unfortunately, institutional addresses are not available right now for some authors.

2#.       Please be aware of the using of specific epidemiological/statistical terms: recommend to use another word instead of “cohort” in Introduction section line 77 since this is a retrospective study and cross-sectional analysis on pathological samples. Recommend to use median instead of mean in abstract section line 30.

Our response / action taken: We acknowledge these comments and have checked and corrected the nomenclature as follows:

The term “set” is used instead of “cohort” in Introduction section line 78

The term “median” is used instead of “mean” as you had recommended, abstract section    line 31

Round  2

Reviewer 1 Report

The authors have appropriately addressed the concerns that were made by the inclusion of a revised discussion, changes to Figure 3 and specific changes to the text. In keeping with these changes, the abstract should include one sentence that reflects the revised text with regards to potential limitations.  Different detection methodologies have unique advantages and limitations and a more comprehensive analysis of this patient set is warranted.

Author Response

Reviewer 1

The authors have appropriately addressed the concerns that were made by the inclusion of a revised discussion, changes to Figure 3 and specific changes to the text. In keeping with these changes, the abstract should include one sentence that reflects the revised text with regards to potential limitations.  Different detection methodologies have unique advantages and limitations and a more comprehensive analysis of this patient set is warranted.

Replies: Thank you for the suggestion. We have included the sentence as recommended and we hope it will be up to your expectation. Please refer to the abstract line 39.

Reviewer 2 Report

Based on the authors feedbacks, as long as they explain in the methods section that the selection/pairing of samples vs. the analysis methods is

largely based on/limited by the availability of the limited pathological

tissues, it will be fine.

Author Response

Reviewer 2

Based on the authors feedbacks, as long as they explain in the methods section that the selection/pairing of samples vs. the analysis methods is largely based on/limited by the availability of the limited pathological tissues, it will be fine.

Replies: Thank you for the comments and positive feedback. A sentence has been added at lines 160-161